# Problem Solving Through Human-AI Preference-Based Cooperation

## Abstract

While there is a widespread belief that artificial general intelligence (AGI) – or even super-human AI – is imminent, complex problems in expert domains are far from being solved. We argue that such problems require human-AI cooperation and that the current state of the art in generative AI is unable to play the role of a reliable partner due to a multitude of shortcomings, including inability to keep track of a complex solution artifact (e.g., a software program), limited support for versatile human preference expression and lack of adapting to human preferences in an interactive setting. To address these challenges, we propose `HAI-Co`$^2$, a novel human-AI co-construction framework. We formalize `HAI-Co`$^2$ and discuss the difficult open research problems that it faces. Finally, we present a case study of `HAI-Co`$^2$ and demonstrate its efficacy compared to monolithic generative AI models.

## 1 Introduction

Despite the impressive achievements of generative AI spearheaded by Large Language Models (LLMs), Vision Language Models and code models (Lozhkov et al., 2024; Wang et al., 2021), multiple recent investigations have pointed out their lack of competence in dealing with complex generation problems that require intricate planning (Kambhampati et al., 2024) or task adherence while keeping track of multiple constraints (Xie et al., 2024). A broad class of such complex problems requires active human participation. Therefore, although the recent focus in generative AI has mostly been on complete automation, we believe that human-AI cooperation is a more promising approach for complex problems of this kind.

To effectively support human-AI cooperation, we draw inspiration from how humans collectively address complex problems to devise solutions: Humans often solve complex problems by iteratively co-constructing a solution step by step, revising and refining draft solutions while transitioning between different levels of abstraction, and exchanging information about preferences and potential improvements in natural language (Damşa, 2013). *Our position is that this type of human-human cooperation is a promising template for the cooperation of humans and AI agents.* On this basis, we propose **H**uman-**AI Co-Co**nstruction (`HAI-Co`$^2$), a novel framework for human-AI cooperative problem solving that builds on preference-based learning and search methodology and relies on natural language to facilitate interaction. The problem solving process is conceived as a process of *systematic search* in a *construction space* $\mathcal{X}$ of *candidate solutions*, i.e., as a co-constructive process, in which candidate solutions are modified step by step until a sufficiently good solution has been found.

**What are co-construction problems?** The broad class of problems that we seek to address in this paper are primarily centered around domain-specific expert applications where the task is to construct a solution that meets specific requirements, e.g., a computer program in software engineering or a machine learning pipeline in automated machine learning (AutoML). We choose the term *co-construction* because of the necessity for the AI agent to work closely with human experts.[1] First, capabilities of current AI systems

---

[1] Not all problems require co-construction; some have unique solutions independent of human feedback. As AI models become more capable, they will be able to solve such problems, which currently require co-construction due to limited AI capability, without help from the human expert since they can better utilize prior knowledge and task specifications. However, in many cases, task descriptions remain inherently incomplete due to either ambiguous specifications or evolving expert preferences, necessitating co-construction with human input regardless of model capability.

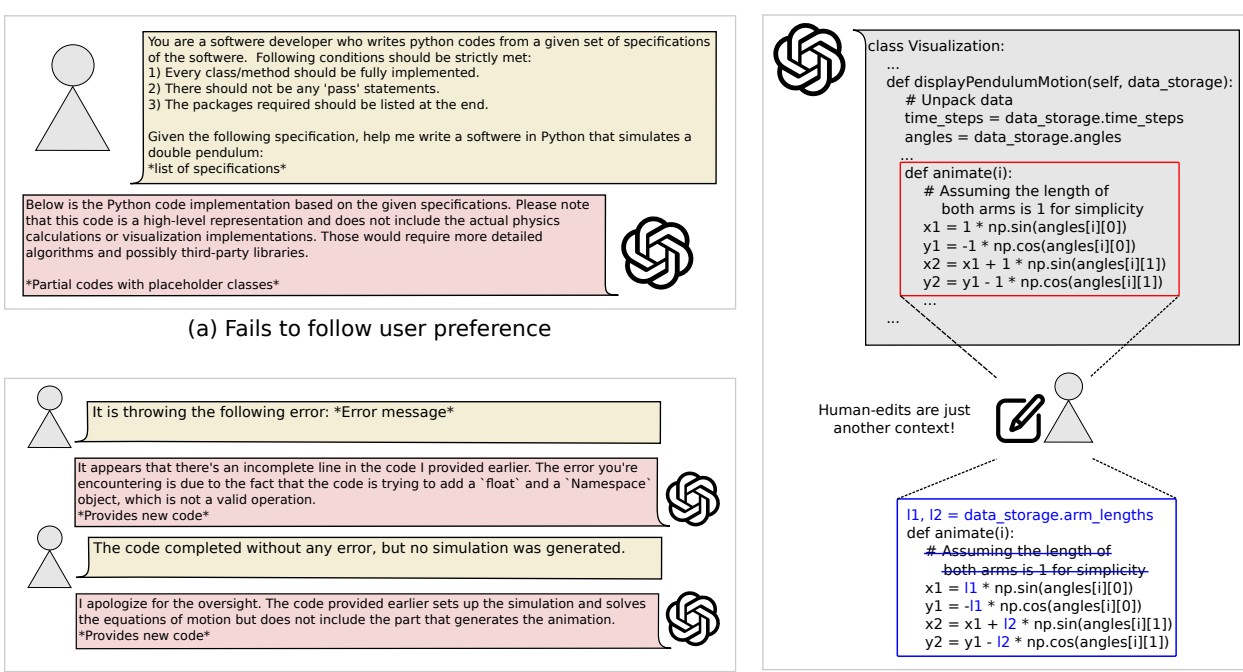

Figure 1: Existing generative AI lacks proficiency in key aspects of co-construction of solutions to complex problems. We give a code synthesis example. (a) GPT-4 Turbo fails to follow preferences explicitly stated by the human expert. (b) Due to the lack of a persistent object representation, a modification request targeted toward one feature of the desired solution leads to the unwanted (and erroneous) modification of another feature. (c) Human expert modifies the generated code directly to remove inline assumptions and introduce general variables; such active participation is not demarcated and recorded by the AI and there is no facility to extract the implicit preference and follow it elsewhere.

are limited and cannot fully replace human expertise in these tasks, e.g., due to insufficient knowledge and reasoning capabilities, lack of trustworthiness, and bias. Scaling generative AI systems, particularly language models, has demonstrated improved performance across a range of tasks, e.g., mathword problems and commonsense reasoning. However, even the most powerful LLMs show a lack of robustness under different semantic perturbations that would not have fooled an otherwise robust reasoning machine (Li et al., 2024). While it is unwise to rule out future improvements, their current limitation certainly calls for interventions beyond scaling. Second, some expert tasks (e.g., designing websites and building data analysis and visualization software) require irreplaceable human supervision; solutions to be constructed in these problems are defined based on personalized human preferences. While the co-constructed solution must fulfill a set of objective correctness criteria that can be verified using symbolic tools without any human intervention, the human expert imposes a broad set of subjective criteria as well. For example, in a software development setup, the final piece of software must be executable, secure, and bug-free; at the same time, every expert developer has their own style – of coding, commenting, modularization, etc. – that should be followed as well. Similarly, an ML pipeline developer may want a suboptimal classifier to save computational cost and go for optimality in a later stage of development. This implies that the AI agent must be able to follow and adapt to the personalized preferences of the human expert (which often are not stable, but evolve in the course of the problem solving process).

Since we see expert domains as the primary setting in which co-construction excels at solving problems, we use "expert" and "human" interchangeably in this paper.

**Is the current state of generative AI enough?** Multiple prior investigations have laid out the inherent shortcomings of present-day generative AI that limit its applicability to the type of problems we are addressing in this paper. In an example case study tailored towards code generation presented in Figure 1,

we demonstrate some of the major bottlenecks of GPT-4 Turbo, a recent update[2] over the original GPT-4 model (OpenAI, 2024). In Figure 1 (a), GPT-4 Turbo ignores the human expert's explicit instructions to generate a complete Python code with the required module specifications, echoing Xie et al. (2024)'s observation that current language-based AI agents lack task adherence. After repeated prompting with partial code snippets, the process produces a complete – albeit faulty – code. This limitation becomes even more irreconcilable as more varied and realistic expressions of human preferences are taken into account – for the human expert to contribute productively, one must allow preferences expressed via explicit instructions, binary choice, ranking, etc. Current generative AI solutions do not facilitate such multi-modal preference incorporation. Figure 1 (b) shows unreliable debugging attempts. Specifically, the LLM performs unrelated (and faulty) edits to address a bug and even introduces new errors. This demonstrates that existing LLMs fail to handle complex, modular software code (Jiang et al., 2024). The common practice is that the human (as a knowledgeable expert who keeps track of overall context) identifies faulty output and repeatedly prompts the model to guide it to the correct generation – this is implicitly adopting a co-construction paradigm. However, Figure 1 (c) shows that current modes of human-AI interaction cannot unleash the full potential of co-construction – direct modification of the co-constructed candidate solution by the human expert does not bear any special significance to the LLM, and it treats it as just another context. There is no explicit mechanism for the AI to learn implicit preferences expressed by the human through active participation.

While these examples are focused on code synthesis, there is evidence of similar shortcomings in other domains (Kambhampati et al., 2024; Lecler et al., 2023; Almarie et al., 2023). Carroll et al. (2019), for example, demonstrate the necessity of incorporating explicit "human-awareness" in a version of the collaborative game Overcooked, providing evidence that agents fail to coordinate with human subjects without such awareness. Code synthesis in particular – and the experience from day-to-day use of generative AI for solving complex problems in general – points toward co-construction as a naturally evolved problem solving paradigm where the human expert tries to search for the optimal solution by interacting with the AI. However, the current state of generative AI hinders its role as a reliable partner in successful co-construction. This is because the "one-directional" interaction between human and AI typical of how AI agents are used today often fails to steer the co-construction towards a solution that satisfies the user's constraints.

**Our contribution.** In this paper, we formalize co-constructive problem solving and thereby aim to address important limitations of current generative AI models. We present $\texttt{HAI-Co}^2$, a conceptual framework with three fundamental properties that facilitate human-AI co-construction. First, it introduces multiple levels of abstractions to the candidate solution, providing a seamless interface for the human expert and the AI agent to modify and keep track of the complex, modular, co-constructed candidate solution. Second, $\texttt{HAI-Co}^2$ allows multi-modal preference input from the human expert, with natural language as the central mediator to capture information-rich guidance signals, along with other forms of active expert participation, such as categorical choice-based preference. Finally, $\texttt{HAI-Co}^2$ introduces a search-based methodology of co-construction where the candidate solution (represented on multiple levels of abstraction) is iteratively revised to maximize the perceived utility modeled from the preference input. While multiple components of $\texttt{HAI-Co}^2$ have been explored in prior research independently across different domains, we are the first to bring them together under a unified conceptual umbrella and to show that jointly with current generative AI, they have the potential to address multiple major challenges in solving expert domain problems. We showcase the efficacy of $\texttt{HAI-Co}^2$ using an example implementation for the expert domain of software engineering.

The rest of the paper is organized as follows. After a discussion of relevant literature in Section 2, we introduce the framework of $\texttt{HAI-Co}^2$ in Section 3 and outline open research challenges implied by this framework in Section 4. Section 5 illustrates $\texttt{HAI-Co}^2$ in a case study, prior to concluding the paper in Section 6.

## 2 Related work

We group relevant prior work into five major strands: reinforcement learning from human feedback (RLHF), assistance games, search- and evolution-driven construction, LLM agents for complex problem solving, and persistent solution spaces for iterative construction. We discuss how these approaches attempt to address

---

[2] We use the `gpt-4-1106-preview` instance: `https://openai.com/index/new-models-and-developer-products-announced-at-devday/`

expert-AI co-construction. However, as we will see in Section 4, they fail to comprehensively tackle these challenges.

**Reinforcement Learning from Human Feedback.** RLHF focuses on learning a policy preferred by humans, most commonly relying on comparisons between candidate solutions (Kaufmann et al., 2024). The goal is to learn a policy that maximizes a reward or utility function that is consistent with the human feedback. Originating in classical reinforcement learning domains such as games and continuous control (Christiano et al., 2017), RLHF has been extended to a variety of domains, most notably fine-tuning generative models such as LLMs (Stiennon et al., 2020; Ouyang et al., 2022), eventually leading to the development of AI models that can generate human-preferred responses in natural language such as ChatGPT.

RLHF for generative AI is typically employed in a single-turn setting, where the agent generates an immediate response to a query, evaluated by a human expert. This contrasts with expert-AI co-construction, which involves multi-turn interactions where agent and expert collaboratively construct a solution. Multi-turn interactions introduce challenges such as extended time horizons and large action spaces. Extensions to RLHF have been proposed that address these issues (Zhou et al., 2024).

Even multi-turn RLHF, however, is not well suited to expert-AI co-construction without further extension: It does not maintain an explicit representation of the solution space, which is crucial for systematic solution construction. In principle, RLHF could be used to learn the AI agent's policy in $\texttt{HAI-Co}^2$, but it is challenging to do so interactively as required in $\texttt{HAI-Co}^2$.

**Assistance Games.** Another approach rooted in reinforcement learning is assistance games (Shah et al., 2021; Laidlaw et al., 2024), originally introduced as cooperative inverse reinforcement learning (Hadfield-Menell et al., 2016). As a framework for cooperative decision-making between an AI agent and a human, assistance games model human-AI interaction as a cooperative game with the agent and the human as players. The AI agent aims to maximize the human's utility function without explicit knowledge of it. It learns this function by observing the human's actions and receiving feedback on its own actions, which may include queries to the human.

The process of co-construction in the sense of $\texttt{HAI-Co}^2$ could in principle be seen as an assistive sequential decision-making problem, too, with the expert's and the agent's actions being steps taken to construct the solution. However, in practice, casting co-constructive problem solving as an assistance game does not seem particularly useful to us. Roughly speaking, for $\texttt{HAI-Co}^2$ we consider co-constructive problem solving as a search rather than a reinforcement learning task. While both tasks are related, RL is a very general setting that comes with additional challenges, notably the learning of a model through exploration (in a non-deterministic environment with unknown effects of actions). Indeed, the price that RL pays for its generality is a high conceptual and computational complexity. This problem applies in particular to assistance games, which, despite their nice theoretical properties (Shah et al., 2021), have not yet been widely applied to real-world problems.[3] We hope that our framework can provide a more practical approach to human-AI co-construction by focusing on the specific challenges of solution construction with a search process guided by the extensive prior knowledge of pretrained language models.

**Learning from natural language interactions.** Natural language interaction between the expert and the AI agent is central to $\texttt{HAI-Co}^2$. A popular approach towards facilitating human-AI interactive problem solving involves training the agent to follow instructions in natural language (Branavan et al., 2009; Tellex et al., 2011). This paradigm of learning to follow instructions has found attention in the LLM era as well (Wei et al., 2022). However, directly mapping language-specified goals to actions has limited applicability to novel tasks. Alternatively, learning rewards from natural language interaction to successfully align the AI agent with the human user has been explored (Fu et al., 2019; Sumers et al., 2021) – instead of learning to map language-defined goals to actions directly, they seek to learn the reward function from the language-defined goals that can be generalized to novel tasks. The findings of Sumers et al. (2022) suggest that while instructions typically perform well in low-autonomy settings, high-autonomy regimes favor the reward-learning paradigm. Instead of learning a language-conditioned policy or reward, it is also possible to use conversational cues as

---

[3]Laidlaw et al. (2024) present a first step towards scalably solving assistance games in more complex environments. This is early work, though, and we believe that the conceptual and computational benefits of the search-based formulation justify a more specialized framework for this highly relevant use case.

rewards themselves (Jaques et al., 2020), which can be combined with the other approaches discussed here. Yet another approach, deployed in OpenAI's ChatGPT, is to enable the language model to save information about the user and their interactions with the model as natural-language 'memories', which may include information about the user's preferences.[4] Most of these approaches do not consider the need to actively elicit and co-construct the expert's preferences, a key aspect of `HAI-Co²`. This necessity is supported by Co-Reyes et al. (2019) and Lin et al. (2022), who identify that inferring the correct behavior or reward from a single utterance can become non-trivial given the multidimensionality of language. Querying the human user and estimating their preferences in an interactive setup is also a key component of Peng et al. (2024)'s framework. As a step in this direction, Li et al. (2023a) use active elicitation to strengthen preference understanding – the AI agent is trained to elicit and infer human preferences by actively interacting with the user. This is crucial prior work for an implementation of `HAI-Co²`, forming an important pillar of future research under the abstract umbrella it provides.

**Search- and Evolution-Driven Construction.** Our framework emphasizes iterative search within the construction space, a process akin to evolutionary optimization, which iteratively generates and evaluates candidate solutions (Bäck, 1996). This evolution can be viewed as a form of search-based construction. Interactive evolutionary computation, a preference-based extension, is particularly relevant to our work as it involves human evaluation of candidate solutions (Takagi, 2001; Wang & Pei, 2024). For example, these methods have been applied to search-based procedural content generation in video games (Togelius et al., 2011). Our approach differs in the core approach to the search process: Traditional evolutionary methods maintain a population of candidate solutions and generate new ones through mutation and recombination. In contrast, in our framework, each iteration ends with a single candidate solution that is then the basis for the next iteration. In addition, we leverage the extensive prior knowledge of pretrained language models to guide the search and use natural language communication to facilitate cooperation between the AI agent and the human expert.

**LLM agents for complex problem solving.** The rapid increase in the capabilities of LLMs has triggered multiple recent efforts to integrate them at the core of autonomous agents that interact with the environment, plan, and act to solve complex problems (Wang et al., 2024). Typical approaches adopt integrating different tool-usage capabilities into LLMs via efficient prompting, often with multimodal capabilities (Chen et al., 2023). A single agent is often insufficient to solve complex problems; thus, multiple agents with different capabilities have to be integrated. Recent efforts in LLM-based multi-agent systems seek to mimic such cooperative problem solving by role-playing LLMs via in-context examples (Li et al., 2023b) or fine-tuning (Juneja et al., 2024). Despite some promising achievements, such agentic ecosystems are inherently limited by the constituent LLMs' inability to plan and execute tasks (Xie et al., 2024; Kambhampati et al., 2024). These frameworks of LLM-based autonomous agents are largely designed for autonomous problem solving, not for human-AI co-construction as we envision. Recently, arguments in favor of strategically allocating tasks between humans and LLM-based agents to exploit their distinct strengths have been put forward (He et al., 2024), which aligns with our approach of leveraging the strengths of both humans and AI agents in co-construction. An alternative view of our framework is hence an extension of LLM-based multi-agent systems with a human agent as a key component, focusing on the co-construction of solutions.

**Persistent Solution Space for Iterative Construction.** A fundamental component of our proposed framework is the explicit representation of the construction space for systematic solution search. Such a persistent memory can be useful for LLM agents. Sumers et al. (2024) propose a cognitive architecture for language agents that connects LLMs to internal memory and external environments, grounding them in existing knowledge or external observations. Similarly, Modarressi et al. (2024) introduce a structured memory component that LLM agents can use for storage and retrieval. In terms of deployed products, Anthropic's artifacts[5] add an explicit representation of an LLM-constructed artifact to the Claude series of language models, which can be iterated on through further interaction. Although these approaches do not directly address expert-AI co-construction challenges, they relate to our approach by providing agents with persistent memory to store intermediate solutions and relevant information for problem solving.

---

[4]`https://openai.com/index/memory-and-new-controls-for-chatgpt/`

[5]`https://support.anthropic.com/en/articles/9487310-what-are-artifacts-and-how-do-i-use-them`

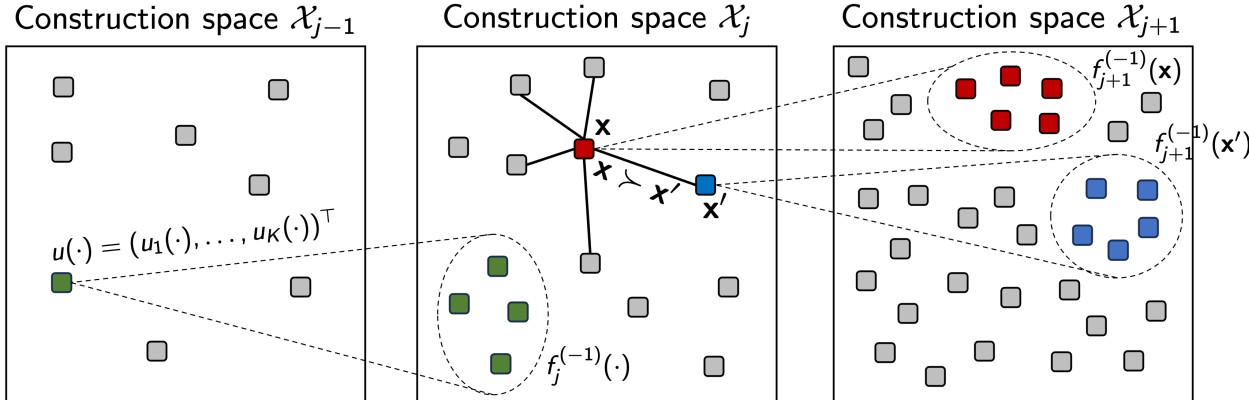

Figure 2: Illustration of the hierarchy of construction spaces in $\texttt{HAI-Co}^2$. Each point $\boldsymbol{x}$ symbolizes a candidate solution (on a certain level of abstraction), e.g., a software program. The topology of the space is specified by a suitable neighborhood structure (as illustrated for point $\boldsymbol{x}$). Each point is associated with a latent utility $u$, possibly multi-dimensional and comprised of local utilities $u_1, \ldots, u_K$, and preferential information (e.g., $\boldsymbol{x} \succ \boldsymbol{x}'$: solution $\boldsymbol{x}$ is better than $\boldsymbol{x}'$) that provides information about promising regions in the space. The relationship between the different abstraction levels is specified by the abstraction mappings $f_j$ resp. the (inverse) refinement mappings $f_j^{(-1)}$.

## 3 $\texttt{HAI-Co}^2$: **Human-AI co-construction through preference-based search**

In this section, we propose $\texttt{HAI-Co}^2$, a framework for cooperative problem solving. Broadly speaking, $\texttt{HAI-Co}^2$ is meant to formalize an interactive problem solving scenario, in which a human expert seeks to (co-)construct a *candidate solution* – such as a computer program – with the help of an AI agent. The problem solving process is conceived as a process of *systematic search* in a space $\mathcal{X}$ of *candidate solutions*, i.e., as a co-constructive process, in which candidate solutions are modified or extended step by step until a (sufficiently) good solution has been found. Therefore, we also refer to the search space $\mathcal{X}$ as the *construction space*. The construction space, its hierarchical organization and its topology (or neighborhood structure) are depicted in Figure 2.

Actions taken by the AI agent during the search (e.g., adapting a candidate solution or asking the expert a question regarding where to move next) depend on its *informational state* $\mathcal{I}$, which comprises its experience so far, e.g., about the expert's preferences, any relevant information about the current context, the solutions considered so far and the best solution constructed so far. Formally, the behavior of the AI agent can be determined by a *policy* $\pi$ that maps informational states to actions.

### 3.1 Construction space and abstraction hierarchy

The construction space will typically be large, most often even (countably) infinite. For example, the construction space may consist of all computer programs in a specific programming language. Spaces of this kind cannot be specified in an explicit way. Instead, they will be defined implicitly and may even be adapted or designed on-the-fly in the course of the problem solving process. In this regard, the *formal representation* of candidate solutions is of major importance and will strongly influence the efficacy and efficiency of the problem solving process. Moreover, it is also clear that the representation of solutions will not be universal but rather specific to the expert domain. For example, a computer program will not be represented in the same way as a machine learning pipeline or data science workflow. It should be noted that we do not make any assumption of *completeness* for candidate solutions: at any stage of the search, a candidate solution $\boldsymbol{x} \in \mathcal{X}$ can be partial or incomplete (i.e., an incomplete codebase, an incomplete ML pipeline, etc.).

During problem solving, it is often useful to look at (candidate) solutions on multiple levels of abstraction. In many cases, for example, a rough draft of the solution is found in a first phase of the process, and this

draft is then worked out in more detail in a second phase. More generally, one can imagine a search process that switches back and forth between different levels of abstraction whenever appropriate. Therefore, we assume the construction space $\mathcal{X}$ is equipped with a hierarchy of abstraction levels.

Formally, this can be modeled by a sequence $\mathcal{X}_0, \mathcal{X}_1, \ldots, \mathcal{X}_J$ of spaces, where $\mathcal{X}_j$ is a refinement of $\mathcal{X}_{j-1}$ – or, vice versa, $\mathcal{X}_{j-1}$ an abstraction of $\mathcal{X}_j$. We describe the abstraction process from $\mathcal{X}_j$ to $\mathcal{X}_{j-1}$ as a surjection $f_j : \mathcal{X}_j \to \mathcal{X}_{j-1}$ such that $\boldsymbol{x}' = f_j(\boldsymbol{x}) \in \mathcal{X}_{j-1}$; that is, $\boldsymbol{x}'$ is the abstraction of $\boldsymbol{x}$ on the abstraction level modeled by $\mathcal{X}_{j-1}$. We denote by $f_j^{(-1)}(\boldsymbol{x}') = \{\boldsymbol{x} \in \mathcal{X}_j \,|\, f_j(\boldsymbol{x}) = \boldsymbol{x}'\}$ the set of all refinements of $\boldsymbol{x}' \in \mathcal{X}_{j-1}$ on abstraction level $\mathcal{X}_j$. Note that refinements are not unique, which is why a transition from $\mathcal{X}_{j-1}$ to $\mathcal{X}_j$ may come with a certain arbitrariness.

In our case study on code generation presented in Section 5, we implement the construction space on three levels of abstraction, considering a Python program as a refinement of a UML diagram, which in turn is a refinement of a natural language specification. The refinement maps are implemented by suitably prompted LLMs that take a representation in a higher abstraction space as input and produce a solution representation in the lower level of abstraction as output.

## 3.2 Latent utility

We assume that the construction space is equipped with a latent *utility function* reflecting the preferences of the expert, i.e., the quality of solutions as perceived by the expert. In general, "quality" may refer to various dimensions or criteria, and different objectives might be pursued at the same time; we formalize this with a *multidimensional* utility function $u(\boldsymbol{x}) = (u_1(\boldsymbol{x}), \ldots, u_K(\boldsymbol{x}))^\top$ comprised of local utility functions $u_i$. For example, a computer program could be rated by average runtime or memory consumption. The local utility functions can be combined into a scalar utility function $U : \mathcal{X} \to \mathbb{R}$ via a suitable aggregation operator.

Various factors influencing the quality of candidate solutions can be distinguished, notably hard and soft constraints. *Hard constraints* refer to (functional) properties that qualify a candidate as a valid solution. For example, a computer program should properly compile and not contain any syntax errors. Even if invalid solutions should normally be considered useless, the abstract notion of utility is flexible enough to distinguish different levels of invalidity. For example, a non-executable computer program may still have a non-zero utility if the error can easily be fixed by the expert. In any case, hard constraints will normally not identify a solution uniquely. For example, there are many computer programs that are functionally equivalent in the sense of having the same input-output behavior. *Soft constraints* refer to criteria that make a solution more or less desirable such as the length of a computer program and its time and memory consumption.

In general, the utility (be it in the form of the multidimensional utility function $u$ or the scalar utility function $U$) is not known to the AI, nor is the expert explicitly aware of it. Rather, this utility is latent and underlies the expert's preference feedback. From this preference feedback, the AI can learn an approximation $\hat{U}$. The AI's goal is then to construct a solution $\boldsymbol{x}^*$ that maximizes $\hat{U}$, or which is at least close to the maximizer, while simultaneously improving the approximation quality of $\hat{U}$. The utility $U$ also induces utilities on higher levels of abstraction. For example, one way to "lift" a utility function from level $\mathcal{X}_j$ to the more abstract level $\mathcal{X}_{j-1}$ is via aggregation: $U(\boldsymbol{x}') = \alpha(\{U(\boldsymbol{x}) \,|\, \boldsymbol{x} \in \mathcal{X}_j, f_j(\boldsymbol{x}) = \boldsymbol{x}'\})$, where $\alpha$ is an appropriate aggregation function (Grabisch et al., 2009).

In our case study, we represent the utilities as user preferences in natural language, extracted from the interaction where both the user and the AI agent can choose between presented options, provide explicit instructions, or actively edit parts of the solution.

## 3.3 Interaction and preference-based search

Search through the construction space is guided by an underlying *search strategy* – in principle, any heuristic search method (properly balancing exploration of the construction space and exploitation of acquired knowledge) may serve as a point of departure. However, in `HAI-Co²`, the search is also interactive and largely controlled by the human-AI cooperation.

To guide the search, human and AI can communicate via natural language; e.g., the AI agent may ask the expert for feedback or explicit advice. Alternatively, the expert may actively intervene, for example by critiquing or modifying a candidate solution. A third type of interaction, particularly important in the context of HAI-Co$^2$, is driven through *preferential feedback*: By informing the AI agent about the quality of candidate solutions, the expert provides hints at presumably more promising (and, likewise, less promising) regions of the construction space, and hence suggests promising "search directions" to the AI agent. To give an example, the expert can compare two competing candidate solutions with each other (e.g., whether a modification has improved a solution or made it worse) and provide this feedback (in natural language) to the AI agent for the next iteration. The AI agent utilizes the feedback to improve its approximation $\hat{U}$ of the latent utility function, which is an important element of its informational state.

In our case study, for example, we implement a preference-based search strategy that identifies promising solutions via a tournament of pairwise comparisons. Besides, we realize a search policy that refines an existing solution guided by the expert's preferences (see Section 5 for details).

The way in which the AI agent and the human expert cooperate with each other is defined in the form of a *protocol*. Among other things, the protocol clarifies the types of queries and responses on the two sides (AI agent and human expert) and the (preference) feedback that can be given by the expert.

In summary, the specification of a concrete *instantiation of* HAI-Co$^2$ includes the following elements:

- (Hierarchical) representation of candidate solutions (domain-specific)

- Structure of construction space $\mathcal{X}$, refinement/abstraction mappings, neighborhood structure

- Search operators (for modification of candidate solutions, refinement, abstraction, etc.) and strategy

- Natural language methods and protocol for cooperation

- Representation of informational states, the AI agent's action space and policy

- Utility: soft/hard constraints, preference relations/predicates (i.e., what type of preferences can in general be expressed, and in which form)

While some of these components can be specified by hand, others could be subject to (machine) learning and data-driven adaptation.

HAI-Co$^2$ comprises a broad variety of human-AI interaction in natural language, as well as categorical choices and active modification of the candidate solution by the human expert. The search policy $\pi$ is designed to generate a (locally) optimal candidate solution based on the immediate as well as historical feedback, thereby adapting to the preference signals from the human expert user. The hierarchical abstraction of the search space facilitates a modular modification of the complex candidate solution. As we will see in the case study (Section 5), HAI-Co$^2$ also allows for incorporating creative components into the generation of candidate solutions, for example through the injection of randomness in the heuristic search process or the refinement of abstract into more concrete solutions.

## 4 Challenges and practical issues

Our characterization of HAI-Co$^2$ implies multiple challenges that need to be addressed to realize co-construction effectively. In the following, we briefly describe these challenges with reference to the current state of research.

**Specification of abstraction hierarchy.** Two core components of HAI-Co$^2$ are (i) the abstraction hierarchy of the construction space and (ii) the neighborhood structure that facilitates preference-based search. A synergistic implementation of (i) and (ii) poses a non-trivial research challenge. In the domain of code generation, Le et al. (2024) propose a modular code generation approach to circumvent this challenge: they generate a chain-of-thought style intermediate description of the subtasks followed by modular codes implementing each of them. Such a hierarchical generation approach can be extended to generalized solution

co-construction. However, relying on purely natural language-based intermediate representations limits the utility of the hierarchy. The choice of abstraction representation is domain-specific: UML descriptions are a suitable abstraction for code generation, but not for an AI scientific assistant. The abstraction specification must adhere to the neighborhood structure on all levels of abstraction; e.g., if two points are neighbors, their abstractions must also be neighbors. This poses additional constraints on the choice of abstraction, the choice of neighborhood structure, and the choice of refinements between two given levels of abstraction. Ideally, all levels of abstraction must be suitable for human preference expression: an "unnatural" choice of abstraction can render the co-construction tiresome for the human expert, negatively affecting productivity.

**Learning from active human edits.** The role of the AI agent as a co-construction partner is central to $\texttt{HAI-Co}^2$. This entails the possibility of active participation from the human expert and the need to learn expert preferences from such participation. Current generative AI lacks the necessary structures of the solution space, primarily represented in its input context, that could delineate the changes introduced by the expert and, subsequently, be the basis for learning from it. $\texttt{HAI-Co}^2$ provides a plausible alternative to "put everything in context" that can solve this challenge, as we argue in the following. Let $\boldsymbol{x} \in \mathcal{X}_j$ and $\boldsymbol{x}' \in \mathcal{X}_j$ be the solution before and after the edit from the human expert. The neighborhood structure imposed by $\texttt{HAI-Co}^2$ on $\mathcal{X}_j$ requires learning the changes in the utility function upon moving from a candidate solution to a neighboring one. If one ensures a vector space structure formed by the utility $\boldsymbol{u}(\boldsymbol{x})$, then the expert preference is equivalent to $\boldsymbol{u}(\boldsymbol{x}') - \boldsymbol{u}(\boldsymbol{x})$. Even without learning to map the solutions to the utility, one can simply seek to learn the mapping from $\boldsymbol{x}' - \boldsymbol{x}$ to expert preferences. Gao et al. (2024) previously showed that learning preferences from such changes is superior to prompting-based methods in terms of aligning LLMs to user edits. However, their experiments are focused on simpler, general-purpose natural language generation tasks. In expert-domain applications where the utility of a solution includes multiple hard constraints (e.g., executability of code) along with stylistic preferences, learning a structured representation of the utility space is essential and a challenge on its own.

**Specification of informational state.** $\texttt{HAI-Co}^2$ utilizes an informational state to keep track of relevant information in the interaction history. Given that the search policy is conditioned on it, an efficient representation of the informational state is a core challenge of $\texttt{HAI-Co}^2$. Typically, such interactive co-constructions are expected to span a long sequence context. While we have observed a significant surge in the context-size of present-day generative AI (e.g., GPT-4 Turbo can handle up to 128K tokens in the input prompt), recent research has questioned the effective usability of such very long context information (Liu et al., 2024). The representation of the informational state needs to be compatible with the abstraction specification of the construction space as well as the choice of how preference signals from the human expert are encoded. This is particularly important as the reflection of any preference signal upon the candidate solution is manifested via the informational state – an unreliable update of the informational state subsequently worsens the solution quality and may result in a divergent search.

**Communication in natural language.** When humans co-construct a solution, communication in natural language plays an important role. Natural language is a powerful and at the same time succinct medium for conveying information. Given the expressivity of natural language, human and AI agent can easily communicate different options of how to improve the current solution, both at a detailed level and in more abstract terms. Similarly, preference learning is facilitated by natural language, since many preferences are easily specified in natural language. The challenge here is that the language capabilities of LLMs have advanced to an impressive level for the general domain, but this does not apply to complex expert domains (Magesh et al., 2024; Hager et al., 2024; Anand et al., 2024).

**Multimodal human-AI interaction.** Natural language-based interaction is not the ideal channel for all types of preferences. Categorical preference can be communicated more simply by pointing towards a better solution. Thus, we would like to incorporate multiple types of preference into $\texttt{HAI-Co}^2$. Similar to deciphering the preference from natural language, different modalities and modeling approaches have their own sets of challenges and require non-trivial research efforts. For example, inferring a preference-based global ranking from pairwise comparisons can be challenging. Popular methods like the Bradley-Terry model (Bradley & Terry, 1952) have their own limitations, such as strong assumptions on the preference structure. Prior literature tackling such hurdles (Mao et al., 2018; Shah et al., 2016) paves the way for research under the umbrella of $\texttt{HAI-Co}^2$. Additionally, incorporation of expert preferences across multiple modalities poses the

challenge of aligning these multiple modes of feedback with each other. For example, the human expert may express the need for a security feature in a software engineering problem explicitly, or they can express it implicitly by choosing a candidate solution that includes the feature over one that does not. The AI needs to extract equivalent preference information in these two scenarios. Contemporary research in recommender systems that deal with modeling user preferences on multiple item modalities (Guo et al., 2018; Xu et al., 2021) can serve as a starting point. However, the relative complexity and nuances of preferences in the case of `HAI-Co²` hinder a trivial extension of recommendation-oriented solutions.

**Dynamic user preferences.** Current techniques of aligning neural AI systems to human preferences, broadly referred to as RLHF (Reinforcement Learning from Human Feedback), typically involve a two-stage process: learning a reward model on preference data followed by fine-tuning a foundation model (often an LLM or a diffusion model) upon reward supervision from the reward model (Kaufmann et al., 2024). This setup is fundamentally limited to static adaptation in the regime of expert-domain co-construction; a single model of human preferences is imitated by the agent that cannot adapt to the personalized preferences of the human expert. In practice, user preferences are dynamic and evolve over time (Franklin et al., 2022). They can also be influenced by and hence be constructed during the elicitation process itself (Lichtenstein & Slovic, 2006). This is a fundamental challenge in co-construction problems, where the AI agent must adapt to the evolving preferences of the human expert. Multi-turn RLHF (Zhou et al., 2024), though it extends the context of preference-adherence to an iterative, conversational regime, cannot solve the challenge of dynamically evolving user preferences. The PAL framework, proposed by Chen et al. (2024), provides a partial solution to our problem via personalized modeling of static human preferences. Unlike traditional policy learning, `HAI-Co²` motivates a reward-free exploration of the solution space (Jin et al., 2020). In-context reinforcement learning can pave the way towards handling dynamic preference signals (Yang et al., 2024; Lee et al., 2024). However, the action space in the scope of `HAI-Co²` overlaps with the generation of multiple hierarchical views of the candidate solution, rendering the problem much harder than existing work on in-context RL. Prior work with LLMs showcases the possibilities of using them as in-context agents, though exploration abilities will need fine-tuning-based interventions (Krishnamurthy et al., 2024).

**Creativity-correctness dilemma.** The specific class of co-construction problems that we seek to address requires creative generation. At the same time, in most expert-domain applications, the solution needs to fulfill objective correctness criteria. With generative models, the two requirements of creativity and correctness become counteractive. Creative generation typically emerges in highly stochastic regimes (e.g., high temperature decoding) (Wang et al., 2023a). However, increased stochasticity carries the risk of hallucination (Aithal et al., 2024). For problems with definite answers, it has already been shown that more robust reasoning can be achieved by stochastic exploration of the generation space and identifying the subset of solutions that are most consistent (Wang et al., 2023b). However, such self-consistency methods are limited to problem classes with definitive answers and cannot be readily applied to the co-construction problems that we characterize in this paper. In `HAI-Co²`, this can be generalized into a broader learning problem of exploration-exploitation trade-off. In the early iterations of co-construction, when the preference input from the human expert is likely to be vague, the AI may bias towards exploration of the construction space in search for a creative solution backbone. As the co-construction proceeds, the human expert fixates on the feature requirements and the AI must refrain from abrupt modifications and build on the preference model developed from the early exploration.

**Evaluation of co-construction techniques.** Due to the personalized and dynamic preferences of the expert and the complexity and modularity of the solution, the evaluation of co-construction is a difficult challenge. We identify multiple dimensions of evaluation that need to be addressed:

- *Quality of the solution* should be evaluated using domain-specific measures; irrespective of the process of co-construction or human preferences, the solution must fulfill some objective criteria of correctness.

- *Preference-adherence* is an essential criterion of the co-construction problem; across multi-iteration co-construction, the generation should closely follow what the human expert asks of it.

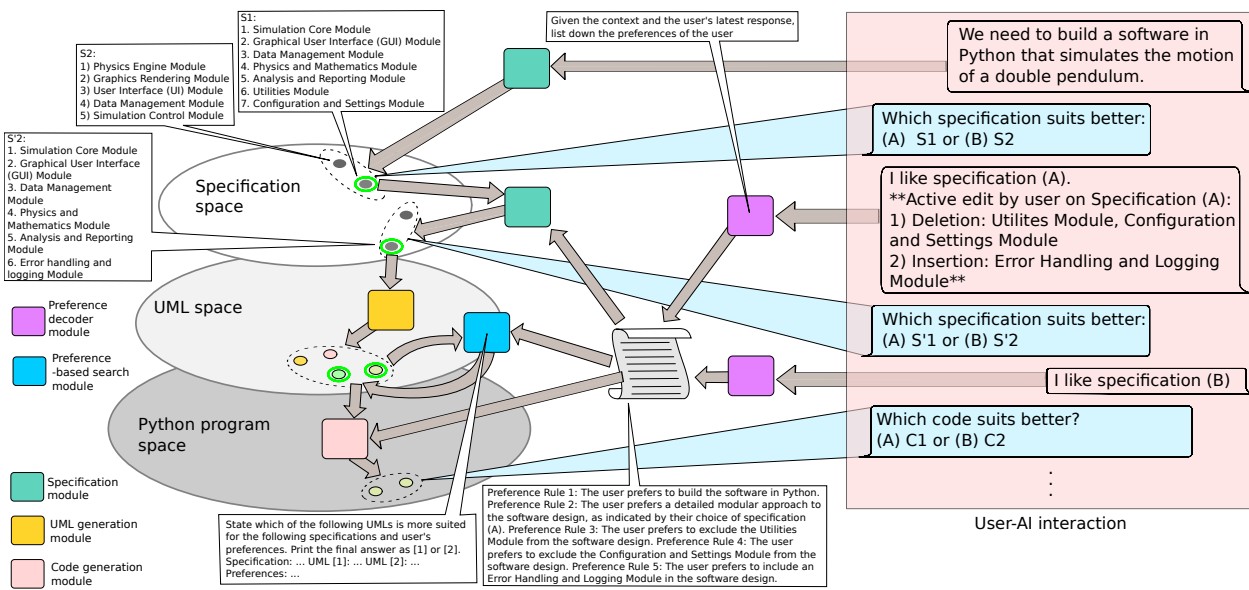

Figure 3: Instantiation of `HAI-Co`[2] for the problem of building a double pendulum simulation. A solution is co-constructed through human-AI cooperation as follows. The interaction between the human and the AI is shown in the red box on the right, top to bottom. We define the co-construction space on three levels of hierarchy: Specification space, UML space, and Python program space. The user starts by specifying the (potentially incomplete) problem to solve. The Specification module (green rectangle) generates a pair of candidate specifications of the software to build (S1 and S2) and presents them to the user. The user expresses their preference in two manners: i) they choose one of the candidate solutions (S1) as better than the other, and ii) provide partial edits to the specification directly. The Preference decoder module (purple rectangle) extracts preference rules from the interaction context. Based on the decoded preferences, the Specification module generates a new pair of candidate specifications (only one shown for space reasons), from which the user chooses one. The UML generation module serves as a generator of refinements from specification space to UML space and generates a set of four UMLs from the specification selected. The Preference-based search module then runs a tournament-based search among these UML candidates: a pair of UMLs are compared against the specification and the decoded user preferences and one is chosen. Two finalist UMLs from the tournament are then used by the Code generation module (pink rectangle) to generate two candidate Python programs. These programs are presented to the user again for their feedback.

- *Self-consistency* is another key aspect of `HAI-Co`[2], as it allows multiple levels of abstraction along with multiple modes of human preference input; it is essential to quantize how consistently the hierarchical abstraction is represented and different modes of preference input are aligned.

- *Complexity* of co-construction includes the computational complexity of generation and preference-based search – resulting in potentially high computational cost – and the cognitive complexity of the framework – resulting in cognitive load for the expert user. The latter demands significant research efforts from a multidisciplinary approach to ensure that automated assistants truly bring value to the expert.

Given that human experts are costly and have limited time, LLM-based simulation of human-AI interaction may facilitate large-scale evaluation (Tamoyan et al., 2024). Even though our four evaluation criteria seem to demand human evaluation, we conjecture that the development of artificial critic models (McAleese et al., 2024), with human value alignment, will be an important research direction in the future.

# 5 An exemplary simulation of `HAI-Co`$^2$

In this section, we provide a case study of `HAI-Co`$^2$, tailored to code generation as a co-construction problem. This case study does not claim scientific rigor on its own; instead, we use it to demonstrate what prior findings (see Section 1 and Section 2) already establish. We do not provide a complete implementation of `HAI-Co`$^2$; in particular, the following are not included in the case study: neighborhood structure of the solution space, preference extraction from actual human participation, dedicated utility function tailored toward the expert problem. Instead, we emulate the intended behavior of a complete implementation using prompted LLMs, with the goal of motivating the practicality of `HAI-Co`$^2$.

The initial problem description is underspecified. During the cooperation, the user can introduce new requirements, ask for modifications to the already generated code, and so on. We approximate different aspects of `HAI-Co`$^2$ (the surjective mappings between different abstraction hierarchies of the construction space, policy and heuristic search strategy) using baseline implementation strategies for ease of demonstration. Future research endeavors should be directed to more in-depth implementation of these features.

**Problem.** The user wants to develop a modular Python codebase for simulating a double pendulum. Modules should include components such as I/O interface, visualization and physics engine.

In this example, the construction space $\mathcal{X}$ consists of the set of all Python programs. Three distinct levels of abstraction are implemented. (i) Specification space. A specification of the simulation software in natural language ($\mathcal{X}_0$). (ii) UML space. A UML description of the software ($\mathcal{X}_1$). (iii) Python program space. The Python program ($\mathcal{X}_2$) itself. The abstraction refinements $f_1^{(-1)}$ and $f_2^{(-1)}$ (as introduced in Section 3) are implemented using suitably prompted instances of GPT-4 Turbo that we denote as *UML generation module* and *Code generation module*, respectively; while the former produces a (stochastic) set of refinements in UML given a natural language specification, the latter generates Python implementations of a given UML description. To decode the user's preferences from the interaction, we use a *Preference decoder module*, implemented using prompted GPT-4 Turbo. Following the focus on natural language, the informational state $\mathcal{I}$ is realized primarily as the interaction history in natural language, along with an explicit list of preference rules decoded from this interaction. One can impose a geometric structure on $\mathcal{I}$ by introducing explicit metric space structure on the different abstraction spaces (e.g., edit distance), rendering $\mathcal{I}$ to behave like a trajectory. However, introduction of such structures will be dependent on the expert domain application.

To facilitate the exploration of the candidate solution space, we generate multiple solution representations on different abstraction levels by setting a high decoding temperature in the respective generation modules and sampling multiple responses. Intuitively, we seek to exploit earlier findings that a highly stochastic generation regime facilitates better novelty (Wang et al., 2023a). Furthermore, this imposed stochasticity can be interpreted as the notion of neighborhood in the respective spaces: one can treat two solutions sampled from the same input context of the generation module as neighbors; distance between two different input contexts can be measured by edit distance. Although we do not explicitly specify such geometric structure in this example, the search strategy uses it implicitly.

We do not implement a concrete realization of the search policy $\pi$; instead, we rely on the limited abilities of LLM instances to explore and implement policy iterations (Krishnamurthy et al., 2024; Brooks et al., 2023). While the notion of search is present across all three levels of abstraction, we perform explicit search in the UML space using the *Preference-based search module*, which runs a tournament among candidate UML solutions, guided by the decoded preferences.

The implementation, as depicted in Figure 3, instantiates `HAI-Co`$^2$ as follows. The co-construction starts with the user providing an underspecified description of the task (in this example, building a simulation software in Python). The specification module (green rectangle in Figure 3) generates the natural language abstraction of the candidate solution as a list of possible components of the software along with their functionalities. This serves as a transparent interface in natural language that provides a layout of the construction. A pair of candidate specifications are generated using a high-temperature stochastic generation regime. The user chooses one of them as better. Additionally, they can state any explicit modification request. Moreover, they can directly edit the specification if they have specific requirements in mind (*preemptive reviewer*), or choose to continue with the workflow and decide on the specifics upon observing the final program (*lazy*

*reviewer*). The Preference decoder module (purple rectangle in Figure 3) lists down the preferences decoded from the user's actions. If the user introduces any new modification (e.g., in the presented example, they remove certain modules from the specification and insert new modules), the specification module generates a new pair of specifications for the user to provide feedback on. This continues till a suitable specification is obtained.

Next, the UML generation module generates a set of stochastic refinements of the natural language specification into UML descriptions. The UML description of the software forces the subsequent code generation module to generate a final program that consists of multiple, independent components (in this case, Python classes) and well-defined dependencies among such components. Micro-level changes to the code (e.g., changing the design of the GUI, choice of numerical algorithms in the simulator, etc.) can be facilitated now without changing the complete codebase – a desirable property of our implementation that is closer to real-life software engineering. This addresses the challenge monolithic code LLMs face in scenarios in which persistent editing is required. However, generating the Python programs from all such candidate UMLs and verifying them one by one is both computationally expensive and infeasible for the human user.

The preference-based search among the candidate UMLs is implemented as a tournament by iteratively declaring one among a pair as the winner of a round. After a logarithmic order of such rounds ($\log_2 n$ being the depth of the tournament tree for $n$ candidate UMLs), the Preference-based search module comes up with a final pair and a summary of preference justifications.[6] Note that although we seek to minimize the cost of human intervention in this step by automating preference-based ranking, one can envision the human expert providing their judgement on these UMLs. In such a setup, the Preference-decoding module can be used to explicitly adapt to such gold preference examples.

Next, we utilize the code generation module to translate each of the two selected UML candidates into a candidate Python program that will be used for human feedback post-execution. Aligned to the goal of co-construction, in this last stage, the user provides their binary judgment on the relative quality of the two generated Python programs along with (optionally) natural language feedback. Such feedback can incorporate the errors found in the program execution (if any), additional requirements, etc. This feedback, along with the summary of the tournament generated by the preference learning module, are together used as a context for the next iteration of refinement. This iterative process continues until the user is satisfied with the solution.

**Comparison with monolithic LLMs.** We perform offline human evaluation to compare `HAI-Co²` against a vanilla LLM (in this case, GPT-4 Turbo) in terms of their effectiveness as co-construction partners for expert domain problems. The problem to be solved is to generate code for the double pendulum simulation. Multiple co-construction episodes are generated by specifying different initial preferences and mid-episode preference switching (e.g., choosing different specifications generated by the Specification module; asking for different functionalities in the software; and editing different segments in the specification as well as the codes). Each human evaluator is presented with a pair of co-construction episodes – one using `HAI-Co²`, one using GPT-4 Turbo – and asked to compare them in terms of the following criteria:

**Q1.** Which assistant has better incorporated the initial preferences of the user?

**Q2.** Which assistant better adapted to preference switching?

**Q3.** Which assistant is more precise in terms of iterative refinement? 'Precision of modification' means that the changes are relevant to the request.

**Q4.** Which assistant is more complete in terms of iterative refinement? 'Completeness of modification' means that all the necessary changes have been made.

**Q5.** Overall, which assistant seems more suitable for software-level code generation?

We recruited a total of 14 participants for this evaluation; each is a doctoral student in Natural Language Processing, so that expert preferences in code generation can be understood. On all five criteria, the majority

---

[6]See `https://anonymous.4open.science/w/ExAIC-Interactions-10A2/PreferenceLayer.html` for the tournament on the candidate UMLs generated

of evaluators rate `HAI-Co`$^2$ higher than GPT-4 Turbo. 12 (85.7%) evaluators find that `HAI-Co`$^2$ better captures the initial preferences of the user (**Q1**). 11 (78.6%) agree that it can adapt better to preference switching (**Q2**). 11 (78.6%) see `HAI-Co`$^2$ as superior on precision (**Q3**) and completeness of modification (**Q4**). 12 (85.7%) suggest `HAI-Co`$^2$ is better suited for software development (**Q5**). We present detailed justifications for the responses provided by the evaluators in Appendix B.

**Limitations and further improvements.** The immediate improvements we observe are prevalent without any dedicated implementation – neither of the two refinement maps (from natural language to UML and from UML to Python) nor of the preference-based search policy. In this implementation, we do not equip the co-construction space with explicit neighborhood structures. We posit that development along these directions will further improve the quality of co-construction and confirm `HAI-Co`$^2$'s potential as an effective framework for the class of co-construction problems we aim to address. Modern software development relies on software engineering tools such as type systems, test drivers, static program analysis tools, monitoring and debugging tools and security vulnerability detectors. Realistic software artifacts are complex, and their full evaluation by humans without these tools is infeasible. Our implementation of `HAI-Co`$^2$– not intended as a systematic evaluation of `HAI-Co`$^2$'s effectiveness in the software domain – will need to be extended with many of the elements that are standard in the DevOps pipeline (see Le et al. (2022); Maninger et al. (2024) for examples of how to integrate such standard tools). We use the LLM's generative capacity as is, e.g., when it explains why one generated solution is better than another. An interesting research direction would be *explanatory interactive learning* (Ross et al., 2017; Teso & Kersting, 2019; Friedrich et al., 2023), where more faithful explanations are produced through interactively constraining model explanations. The search strategy can be further refined by implementing reinforcement learning from execution feedback (Gehring et al., 2024; Liu et al., 2023; Dutta et al., 2024).

## 6 Conclusion

This paper presents a novel research perspective towards complex problem solving via human-AI co-construction. Our position is that existing generative AI agents require active human participation to successfully construct solutions to complex problems, but cannot effectively serve as reliable partners in human-AI cooperation due to their current limitations. We find evidence for this position in multiple prior research areas across a broad set of domains. Our case study focuses on software generation using GPT-4 Turbo, a strong proprietary LLM, and exemplifies the major drawbacks of current LLMs such as inability to follow human preferences, unreliable refinement of complex solution artifacts and limitations to facilitate active human participation. We observed that although day-to-day usage of generative AI tends to adopt a type of human-AI co-construction paradigm in an uninformed manner, the challenges that LLMs face confine such interactions to a much weaker form. As a remedy, we introduce `HAI-Co`$^2$, a framework that is motivated by the effectiveness of collective human problem solving. `HAI-Co`$^2$ facilitates a solution construction space with multiple levels of abstractions, in which human and AI iteratively refine the candidate solution through search guided by human preference. `HAI-Co`$^2$ allows active human participation along with versatility in preference expression. After presenting a formalization of `HAI-Co`$^2$, we discussed the research challenges for this new approach as well as possible future directions for addressing them. Finally, we presented a case study for the application of code generation and noted clear improvements compared to monolithic LLMs.

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

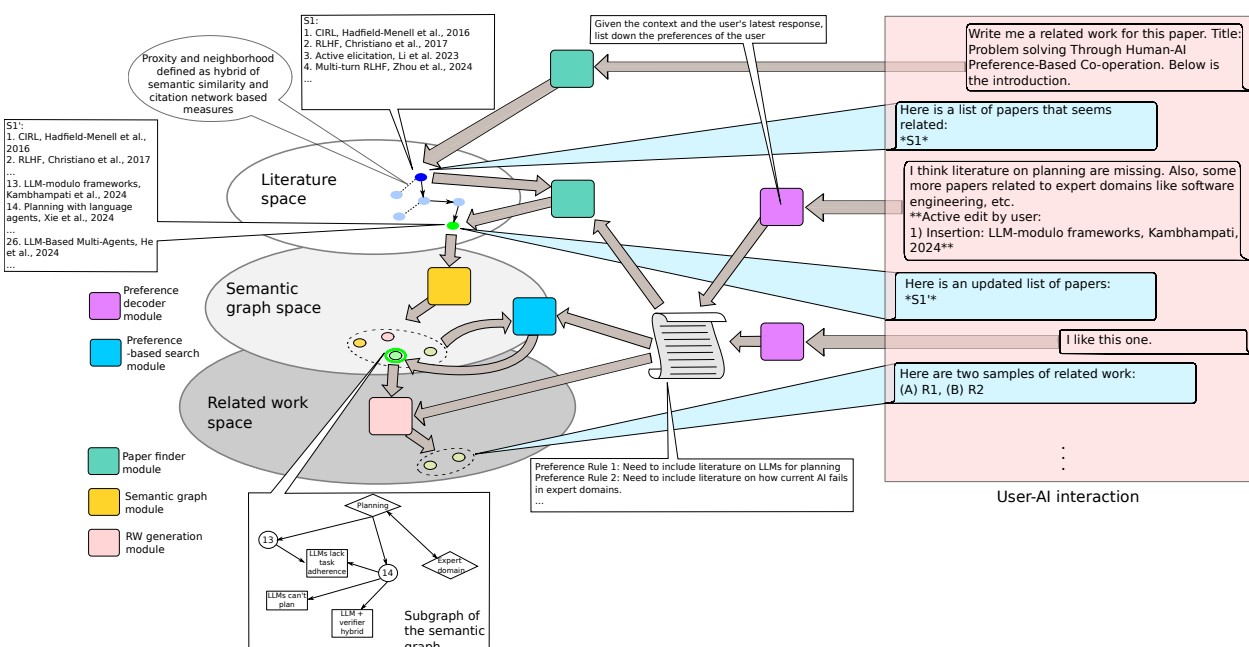

Figure 4: A conceptual application of `HAI-Co`$^2$ for the problem of generation of a related work section of a research paper. The user provides the title and the introduction of the research paper for which the related work section is to be co-constructed (i.e., written). Three abstraction hierarchies are envisioned. The Literature space consists of lists of papers. The Semantic graph space depicts the papers, their findings, and their interrelations using a directed graph. The Related work space contains related work sections (written in natural language using a writing style appropriate for this genre) that describe the semantic graph. The Paper finder module lists down the papers that should be incorporated into the related work. Neighborhood structure is imposed upon this space using semantic similarity and hops over the citation network. The Preference decoding module keeps track of the user's preferences. Upon deciding on a suitable list of papers, the Semantic graph module translates them into a semantic graph. Finally, the Related Work (RW) generation module writes related work sections based on the semantic graph.

Yifei Zhou, Andrea Zanette, Jiayi Pan, Sergey Levine, and Aviral Kumar. ArCHer: Training language model agents via hierarchical multi-turn RL. In *Proceedings of the 41st International Conference on Machine Learning*, volume 235 of *Proceedings of Machine Learning Research*, pp. 62178–62209. PMLR, 21–27 Jul 2024. URL `https://proceedings.mlr.press/v235/zhou24t.html`.

## A Possible implementation in Related Work Generation

To showcase the applicability of `HAI-Co`$^2$ for expert domains other than software engineering, we present a workflow for the problem of related work generation in Figure 4. Note that this is not an actual implementation, rather a proposal on how `HAI-Co`$^2$ can be adapted for this problem. Similar to our case study on software engineering, a solution construction space with three levels of hierarchy is defined. The highest level of abstraction (Literature space) is the space of lists of relevant papers; each point (represented as a list of papers) is intended to capture the literature relevant to the research paper (in this case "Problem solving through Human-AI Preference-Based Co-operation"). Based on the user's problem specification, the Paper finder module (which can be an LLM-web search hybrid) lists the possible papers that are relevant to the research paper. This is a classical search problem. Next, these papers are used to construct a semantic graph that relates different papers according to their domain of focus, methodology, findings, prescriptions, etc. Such a graph is inherently heterogeneous. Multiple semantic graphs can be generated from a given list of papers. One can define a neighborhood over the space of these semantic graphs via edit distance. The search strategy, in this case, can again be realized through a tournament (as in the software engineering

Figure 5: Interface used for human evaluation.

domain presented in Section 5) or through another mechanism (e.g., a specialized module that evaluates semantic graphs based on their graph-theoretical properties). Finally, the RW (related work) generation module translates these semantic graphs into Related work sections. Locally valid neighborhood structures can be constructed using the neural representations of textual differences. The Preference decoder module extracts the preferences expressed by the user to guide the search in different spaces. In this example, one can define refinement maps between the different abstraction levels in a straightforward manner: The semantic graph can be mapped to the list of papers directly as the former has nodes that are members of the latter. Similarly, each pair of nodes and their connecting edge in the semantic graph can be translated to a sentence in the related work section in the Related work space.

# B   Human evaluation

We provide each evaluator with a pair of interactions between i) a user and GPT-4 Turbo [7] and ii) a user and HAI-Co$^2$ [8]. We do not collect any personal information from the evaluators. Detailed responses are available here: https://anonymous.4open.science/w/ExAIC-Interactions-10A2/FormResponses.html.

---

[7]Example interaction can be found at https://anonymous.4open.science/w/ExAIC-Interactions-10A2/Assistant2.html
[8]Example interaction can be found at https://anonymous.4open.science/w/ExAIC-Interactions-10A2/Assistant1.html

