# OpenReview forum: "Problem Solving Through Human-AI Preference-Based Cooperation"
_TMLR — Rejected by TMLR_

### Review · Reviewer_B2w8 · 2024-09-22

**Summary Of Contributions:**

This paper introduces HAI-Co^2, a framework for human-AI collaboration. The framework is motivated by limitations of the interaction capabilities of current AI assistants like ChatGPT: they struggle to learn from corrections, they cannot plan over multi-turn interactions, they often fail to recognize implicit preference feedback, etc. HAI-Co^2 is structured around a search for a solution across multiple abstraction levels. The search involves preference-based feedback to find solutions that satisfy a user's preferences. The authors demonstrate an implementation of HAI-Co^2 for constructing a Python program.

**Audience:**

No

**Broader Impact Concerns:**

I do not believe there are broader impact concerns that need to be addressed.

**Claims And Evidence:**

No

**Requested Changes:**

In light of the weaknesses above, the changes that I believe would raise this to my bar for acceptance could include:

* More precisely formalizing HAI-Co^2 and describing its contribution beyond previous frameworks for human-AI collaboration.
* Clarifying the example implementation's relationship to the formal description of HAI-Co^2.
* More thoroughly evaluating the example implementation.

**Strengths And Weaknesses:**

## Strengths

This paper does a great job of addressing the weaknesses of current AI assistants that it tackles. The examples in Figure 1 all motivate the need for better frameworks for human-AI interaction. The literature review is thorough and further motivates the proposed framework.

## Weaknesses

I will discuss the weaknesses of the paper with respect to the two acceptance criteria for TMLR: support for claims and interest to TMLR's audience. I believe both that the claims made by the paper are not well-supported and that its contribution is thus not enough to be of significant interest.

The main contribution of the paper is the the formalization of HAI-Co^2, which includes an example instantiation. However, the description of HAI-Co^2 given in Section 3 is quite vague. Some parts of the framework are precisely specified (the levels of abstraction and the multidimensional latent utility) but are already present in the literature. Other parts are left unspecified, e.g., the method by which the assistant uses feedback to update its approximation of the utility function, the search operations/policy, etc. While the separation of these components could be a contribution in and of itself, the components are already standard parts of multi-agent/collaborative AI frameworks: policies, action spaces, utilities, constraints, preference feedback, information states, search, etc.

Since the formalization of HAI-Co^2 in Section 3 is vague, the example instantiation in Section 5 could help to clarify the framework. However, the example implementation seems to be missing several of the components specified in Section 3: soft/hard constraints, multidimensional utility, neighborhood structure that satisfies "if two points are neighbors, their abstractions must also be neighbors," etc. Furthermore, it is unclear if the instantiation actually addresses many of the problems with existing assistants that the authors claim to tackle in the introduction. The authors only provide anecdotal evidence or heuristic arguments that HAI-Co^2 is more effective than LLM chatbot assistants. Only a single coding problem is considered and no quantitative evidence is given. This undermines the authors' argument that HAI-Co^2 is superior to current assistance approaches.

Overall, it is unclear what the contributions of the paper are and this makes it unclear if it would be of interest to the TMLR readership. One possible contribution is formalization of HAI-Co^2, but it is only vaguely described in terms that do not significantly build beyond current approaches to assistance in the literature. Another is the demonstration of HAI-Co^2's efficacy through the example implementation, but the example is unclearly linked to the formal description and its evaluation is extremely limited.

## Small issues/typos:
 * Page 7 "Wwo core components" -> "Two core components"

---

> ### Author Response · Authors · 2024-11-08
> **Response to reviewer B2w8**
>
> We thank the anonymous reviewer for their valuable comments. We have tried our best to revise the manuscript based on their suggested changes (marked in blue in the revised manuscript). In what follows, we explain the changes in the light of our rationale.
>
> ## Clarification on contribution
>
> We agree that multiple components of HAI-Co2 have been explored in prior research independently across different domains and we do not claim our novelty in terms of them. Instead, we are first to bring them together under a unified conceptual umbrella and show that together with current generative AI, they have the potential to address multiple major challenges in solving expert domain problems. The goal of our paper is to 1) establish the need for human-AI coconstruction, 2) identify the challenges of coconstruction using existing generative AI, 3) identify possible solutions in existing research that can tackle different aspects of such challenges, 4) design a unified framework to bring together these components into the domain of expert problems, and 5) identify what is yet to be done. We have updated the contribution statement in the revised manuscript under Section 1 accordingly to emphasize this. Furthermore, we have updated Section 4 to show how learning from active human edits is facilitated under the neighborhood structure and possible directions of learning from language-based interactions.
>
> ## Formalization of the framework
>
> Some parts of the framework, including the method for preference learning (by which the assistant uses feedback to update its approximation of the utility function) as well as the search policy, are very application-specific, and we don't see how these components can be meaningfully described in a more formal way on the level of a framework (rather than a concrete method).
>
> Please note that other frameworks are not described in more technical detail either. In reinforcement learning, for instance, there are many ways to model states, to learn a policy, etc. Again, these components are application-specific, and concretely described only on the application level.
>
> ## Limitations and underspecification of the example case study
>
> We agree that the implementation of HAI-Co^2 is incomplete. Instead of an actual implementation of the underlying principles of HAI-Co^2 (e.g., neighborhood structure of the solution space, preference extraction from actual human participation, dedicated utility function tailored toward the expert problem, etc.), this implementation seeks to emulate the intended behavior using prompted LLMs, motivating the practicality of HAI-Co^2. We have reiterated this in the revised manuscript under Section 5. In the revised version, we have incorporated i) a Preference-decoder module that extracts and lists expert preferences that are used in the search across different levels of hierarchy, ii) the ability for the user to actively edit the solution, and for the AI to learn the preferences from such edits. We have further elaborated on how this implementation treats the informational state, and how the neighborhood structure can be interpreted using the stochastic decoding and edit distance between two solutions. Despite the incompleteness, a fundamental difference in terms of preference adherence can be observed. The GPT-driven search (which can be used in itself or in conjunction with human judgment) lets the UML generator explore the solution space while lessening the dependence on instruction following and selects the best course among many. Subsequently, it requires far less direct human engagement (by writing a prompt and asking the model to correct something) in this setup compared to vanilla LLMs.
>
> ## Quantitative evaluation of HAI-Co2
>
> We have implemented an offline human evaluation comparing the abilities of HAI-Co2 with that of the vanilla LLM (in this case, GPT-4 Turbo). Each evaluator was presented with a pair of user-AI interaction histories using the two methods and asked to compare them against different metrics. We have included the details and the results of the evaluation under Section 5 and Appendix B in the revised manuscript.

---

### Review · Reviewer_MrCh · 2024-09-24

**Summary Of Contributions:**

This paper offers a critical take on the standard human-LLM chatbot dynamic, which linearly fills up the LLM's context window with interactive context. The authors highlight concerns about the lack of "learning" during this interaction, and corresponding limitations for human-AI collaboration. They propose a comprehensive and generic framework, HAI-Co^2,  which formally describes an iterative process of aligning on a desired solution artifact. They instantiate a limited version of said framework and claim several advantages over the typical linear interaction.

**Audience:**

Yes

**Broader Impact Concerns:**

None.

**Claims And Evidence:**

No

**Requested Changes:**

**Clarifying the scope of HAI-Co^2 [would strengthen the work].** The authors repeatedly make claims that human interaction is required for complex problem solving and reference "a broad class of... complex problems." This claim strikes me as underspecified and overly general. Many complex problems in expert domains do not, in fact, appear to require human-AI cooperation (AlphaGo; or SWE-Bench; or other domains where solution correctness can be verified programmatically). Even in cases where the human expert has style preferences, it may be possible to address these without interaction by simply providing a style guide to the LLM. I think this is a minor weakness that can be addressed by narrowing to claims to more specific domains where HAI-Co^2 will be most useful  to include nuances such as (1) when an expert is uncertain over their preferences at the start and (2) consideration of when HAI-Co^2 is helpful with today's LLMs vs future, stronger ones.

**Grounding claims in prior work [critical for recommendation of acceptance].** HAI-Co^2 shares many similarities existing approaches to interactive preference learning; I think the authors do themselves a disservice by not engaging more deeply with this literature. I provided a list of relevant work I'm aware of in "Strengths and Weaknesses"; I would strongly encourage the authors to take a look and use them as a starting point to explore this area more deeply. I believe that clarifying how HAI-Co^2 relates to these approaches will both sharpen the contribution and engage the relevant communities. Relatedly, it might be helpful to compare HAI-Co^2 to the scaffolding built in top of existing chatbots -- for example, ChatGPT's "memory" feature, or Anthropic's "artifacts" -- which, by my read, appear to be partial instantiations of the ideas discussed here.

**Evidence for benefits of HAI-Co^2 [critical for recommendation of acceptance].** I would like to see quantitative evidence that HAI-Co^2 offers a benefit over a naive LLM interactions, using human participants at a reasonable scale, with a particular emphasis on how multiple rounds of refinement are necessary to accomplish a task. I view this as "table stakes" to substantiate the claims that HAI-Co^2 is an improvement over existing approaches and justify the additional complexity.

As an alternative to actual empirical validation, it might be possible to reframe the paper as a purely conceptual contribution. If the authors were to pursue this route, I think it would require tempering many of the claims and more deeply grounding HAI-Co^2 into the existing literature on preference learning; but it could ultimately produce an interesting paper which brings those works into the LLM era and provides a roadmap for future research.

**Strengths And Weaknesses:**

**Strengths:**

- **The formal framework is intuitively appealing.** Several aspects -- such as the hierarchical structure, importance of both explicit and implicit feedback, and iterative nature of preference elicitation -- are important additions to collaborative interactions. The idea of hierarchical structure, while difficult to realize in practice, feels particularly helpful given humans' ability to express levels at varying levels of abstraction. I appreciated the authors' thoughtfulness in this design.
- **The roadmap for future work is valuable.** The authors engage deeply with numerous limitations of the existing interaction with LLMs, as well as their proposal. I found this guidance interesting and think it could be of value to direct efforts in the HCI-LLM community.

**Weaknesses:**
- **The scope of HAI-Co^2 is underspecified.** A clearer line should be drawn between the types of interactions that (1) LLMs can handle today without collaboration; (2) LLMs can handle in principle without collaboration; and (3) LLMs cannot and will not be able to handle without active collaboration. In general, methods that scaffold around LLMs tend to become irrelevant when LLM capabilities improve; I think it would significantly clarify the claims and overall contribution if the authors are more precise about the kind of interactions that HAI-Co^2 will remain relevant for (as opposed to the ones that is is presently helpful for).

- **Claims are not sufficiently grounded in prior work.** The authors mention but do not engage deeply with the "Assistance games" literature, which I think is highly relevant (and potentially even formally equivalent to?) HAI-Co^2. In addition to some highly relevant work on interactive collaboration with language models [Li et al below], the general literature on reward learning from interactions contains many examples where humans can provide language, demonstrations, or some mix of behaviors and the agent learns their preferences. Many of these approaches are designed to handle the same problem as HAI-Co^2: modeling how some mix of language and behaviors reveal the human's implicit preferences, and adapting the agent's behavior accordingly.
The authors mention this general area but do not compare the specific methods to HAI-Co^2, suggesting that their approach is "more practical." I feel this claim requires further substantiation, given that:
1. Formally, HAI-Co^2 feels similar to prior methods in this literature (a preference-guided search over artifact outputs, as opposed to agent policies).
2. The authors note that HAI-Co^2 is very difficult to implement (i.e. requiring neighbor-preserving hierarchical representations of the solution space), and test only a reduced version of the framework.
Therefore it is not clear to me what HAI-Co^2 is really contributing relative to his existing literature. Please see the next section for some suggestions on how to address this weakness.

- **Claims around the benefits of HAI-Co^2 are not supported.** The implementation of HAI-Co^2 (Section 5) doesn't address fully address the challenges presented by the formal framework (such as ensuring consistent neighborhood structure between levels of abstraction). Further, the implementation is really a partial implementation of HAI-Co^2 and a qualitative discussion of differences. This does not, in my mind, adequately support the author's claims that HAI-Co^2 is a practical method for improving human-LLM interactions. Please see the next section for some suggestions.

--------

Other notes:

**Typo.** "Wwo" on p. 7.

**Broken resources.** I tried to access the anonymized Github repositories linked in the paper but was not able to (I got Cloudflare errors).

**Specific literature.** I'm not asking the authors to necessarily discuss all of these (which do include some of my own work), but I think that many of them (in addition to others I haven't listed) are highly relevant and have immediate bearing on this paper.
- The most relevant work I'm aware of: Li, Belinda Z., et al. "Eliciting human preferences with language models." arXiv preprint arXiv:2310.11589 (2023).
- On learning preferences from natural language interactions
	- Maclin, Richard, and Jude W. Shavlik. Incorporating advice into agents that learn from reinforcements. University of Wisconsin-Madison. Computer Sciences Department, 1994.
	- Fu, Justin, et al. "From language to goals: Inverse reinforcement learning for vision-based instruction following." arXiv preprint arXiv:1902.07742 (2019).
	- Sumers, Theodore R., et al. "Learning rewards from linguistic feedback." Proceedings of the AAAI Conference on Artificial Intelligence. Vol. 35. No. 7. 2021.
	- Lin, Jessy, et al. "Inferring Rewards from Language in Context." Proceedings of the 60th Annual Meeting of the Association for Computational Linguistics (Volume 1: Long Papers). 2022.
	- Sumers, Theodore, et al. "How to talk so AI will learn: Instructions, descriptions, and autonomy." Advances in neural information processing systems 35 (2022): 34762-34775.
	- Peng, Andi, et al. "Preference-Conditioned Language-Guided Abstraction." Proceedings of the 2024 ACM/IEEE International Conference on Human-Robot Interaction. 2024.
	- Jaques, Natasha, et al. "Human-centric dialog training via offline reinforcement learning." Proceedings of the 2020 Conference on Empirical Methods in Natural Language Processing (EMNLP). 2020.
- More general human-AI interaction literature
	- Co-Reyes, John D., et al. "Guiding policies with language via meta-learning." arXiv preprint arXiv:1811.07882 (2018).
	- Carroll, Micah, et al. "On the utility of learning about humans for human-ai coordination." Advances in neural information processing systems 32 (2019).
	- Knox, W. Bradley, and Peter Stone. "Interactively shaping agents via human reinforcement: The TAMER framework." Proceedings of the fifth international conference on Knowledge capture. 2009.
	- Jeon, Hong Jun, Smitha Milli, and Anca Dragan. "Reward-rational (implicit) choice: A unifying formalism for reward learning." Advances in Neural Information Processing Systems 33 (2020): 4415-4426.
- Coordinating on conventions in humans
	- McCarthy, William P., et al. "Learning to communicate about shared procedural abstractions." Proceedings of the Annual Meeting of the Cognitive Science Society. Vol. 43. No. 43. 2021.

---

> ### Author Response · Authors · 2024-11-08
> **Response to reviewer MrCh**
>
> We thank the anonymous reviewer for their valuable comments. We have tried our best to revise the manuscript based on their suggested changes (marked in blue in the revised manuscript). In what follows, we explain the changes in the light of our rationale.
>
> ## Scope of HAI-Co2
>
> Thank you for pointing this out. We have updated the discussion on the kind of problems we envision to be in the purview of our framework. Note that 'what kind of problems can current LLMs solve' is a rather tricky question. Benchmarks that were earlier thought to be solved (e.g., GSM8K) have now been shown to be questionable in terms of robustness. It is hard to say whether LLMs on their own can solve mathword problems or not. Subsequently, it is hard to predict if future generative AI can solve them or not in a generalizable manner. However, we stress that there are certain problems where additional symbolic verifiers can verify the correctness of the solution and therefore, one can expect a 'correct' solution after an arbitrary amount of iterative refinement between the LLM and the symbolic verifier. However, in many expert domain problems, there can be many 'correct' solutions but a few 'desired' ones. The latter requires human intervention. We have incorporated an elaborate discussion in Section 1 in the revised manuscript.
>
> ## Grounding in prior research
>
> In the revised manuscript, we have incorporated discussion around prior research missing earlier and its relationship to different components of HAI-Co2. We have updated the discussion around learning from natural language based interactions in Section 2. We have positioned our research against assistance games with more in-depth engagement in Section 2. We have pointed out how learning from active human edits can be connected to the neighborhood structure proposed in Section 3 and how recent research actually performs an implicit approximation.
>
> ## Limitations and underspecification of the example case study
>
> We agree that the implementation of HAI-Co^2 is incomplete. Instead of an actual implementation of the underlying principles of HAI-Co^2 (e.g., neighborhood structure of the solution space, preference extraction from actual human participation, dedicated utility function tailored toward the expert problem, etc.), this implementation seeks to emulate the intended behavior using prompted LLMs, motivating the practicality of HAI-Co^2. We have reiterated this in the revised manuscript under Section 5. In the revised version, we have incorporated i) a Preference-decoder module that extracts and lists expert preferences that are used in the search across different levels of hierarchy, ii) the ability for the user to actively edit the solution, and for the AI to learn the preferences from such edits. We have further elaborated on how this implementation treats the informational state, and how the neighborhood structure can be interpreted using the stochastic decoding and edit distance between two solutions. Despite the incompleteness, a fundamental difference in terms of preference adherence can be observed. The GPT-driven search (which can be used in itself or in conjunction with human judgment) lets the UML generator explore the solution space while lessening the dependence on instruction following and selects the best course among many. Subsequently, it requires far less direct human engagement (by writing a prompt and asking the model to correct something) in this setup compared to vanilla LLMs.
>
> ## Quantitative evaluation of HAI-Co2
>
> We have implemented an offline human evaluation comparing the abilities of HAI-Co2 with that of the vanilla LLM (in this case, GPT-4 Turbo). Each evaluator was presented with a pair of user-AI interaction histories using the two methods and asked to compare them against different metrics. We have included the details and the results of the evaluation under Section 5 and Appendix B in the revised manuscript.

---

### Review · Reviewer_ucrj · 2024-10-08

**Summary Of Contributions:**

This paper identifies several drawbacks in using LLMs to aid human experts to find satisfactory solutions to complex generation problems in an interactive manner. Motivated by these drawbacks, authors propose a framework named HAI-Co^2, drawing insights from the way in which human approach complex problems. This framework facilitates better cooperation between LLMs and human experts to construct good solutions, especially in incorporating multi-round and iterative derivation. The framework requires that the construction space (space in which agents search for solutions) has a hierarchical structure. This framework also utilizes various forms of feedback from human experts and rely on the search ability of LLMs agents to “converge” to a good solution. Authors provide a case study in which they illustrate how HAI-Co^2 can be implemented for the task of code generation in Python.

**Audience:**

Yes

**Claims And Evidence:**

Yes

**Requested Changes:**

1. The conclusion section states that authors find evidence of the incompetence of generative AI agents in multiple domains, but in Section 1, only examples on code synthesis are given. It would be nice to verify the identified shortcomings in other tasks/domains, to support that they are indeed general.
2. In Section 5, the comparison with monolithic LLMs is summarized into three parts. It would be nice to see some more quantitative, or at least clearer comparisons to support these advantages of the proposed framework, e.g. for preference adherence. Perhaps the most basic visualization is to highlight parts where monolithic LLMs failed to follow expert preference while the proposed framework generated results adhering to the same preference.

Also there are some typos that needs to be fixed, e.g., “ Wwo core components” on page 7.

**Strengths And Weaknesses:**

I do not actively work with the field of generative AI / LLMs so my reviews wrt this part should be taken with a grain of salt.

*Strengths:*
1. The writing is mostly good, and the flow of the paper is easy to follow. Authors clearly establish the expectations at the beginning of each section.
2. Paper presents a nice perspective highlighting some drawbacks of using LLMs to aid human experts in constructing complex solutions, and a thorough discussion of possible fixes.
3. The proposed pipeline for better facilitating human and LLM cooperation for code generation is very relevant right now. The proposed framework has the potential to inspire other interesting future works.
4. The connection to related works is established clearly.

*Weakness:*

I found that the descriptions in this paper tend to stay at a relatively high level without delving into concrete solutions. The description of HAI-Co^2 is in some sense too general and lacks specificity in some key components, making it difficult to see how to actually apply it to address a complex problem at hand (maybe besides the example in Section 5).  This is my main concern with the paper, for more details, see points below.
1. It is claimed that HAI-Co^2 incorporates active modification of the candidate solution by the human expert (ref: last paragraph in Section 3), however, it is unclear how such active modifications can be incorporated in a way that is substantially different than a context. I think it is important to at least provide an example of this, especially given how various forms of human feedback is handled in the exemplary implementation of HAI-CO^2 for code generation in Section 5. (Please see point 2.)
2. In Section 5, various forms of feedback including binary comparison preference from human expert; errors found in the final candidate solution; additional requirements described in natural language, are all handled as *context* for the UML generation module (which itself is an instance of GPT4 Turbo). This seems to defeat the purpose of this paper in some sense, since one of the motivations for deriving a better framework to facilitate generative AI and human cooperation is, as claimed, “current modes of human-AI interaction cannot unleash the full potential of co-construction – direct modification of the co-constructed candidate solution by the human expert does not bear any special significance to the LLM and it treats it as just another context” / in Figure 1 (c), this drawback is also highlighted “Human-edits are just another context” – it is not clear to me what has changed in the proposed framework regarding this drawback.
3. Utility function: In Section 3, it seems the key assumption for HAI-Co^2 is that there exists a latent utility function that the AI agent aims to approximate (as $\hat{U}$) and maximize (“construct a solution $x^*$ that maximizes $\hat{U}$”). The utility function is not defined nor mentioned in the realization of HAI-Co^2 in Section 5. What would the learning process of this utility function be?
4. The assumption that the local utility functions are aggregated into a single utility function U which outputs a scalar, and that the AI agent will aim to optimize U seems restricting. In many complex problems, there are many criterias for a good solution, creating a sense of Pareto optimality for solutions.
6. Given that the instantiation of HAI-Co^2 for code generation asks for binary (pairwise comparison) feedback from experts many times, perhaps a follow up question is: A prominent problem in the literature of learning from pairwise comparisons is how to handle the noise in the human preference feedback (see some reference at the end of this part). In the case where human experts makes an error in their binary preference feedback, or if there is disagreement caused by subjectiveness (expert A says solution 1 $\succ$ solution 2, while expert B says solution 1 $\prec$ solution 2), would the proposed framework handle them?
7. Ambiguity regarding the informational state. The informational state component in the proposed framework seems crucially important, according to analysis in Section 4. However, in Section 5, the informational state is described in an unsubstantiated way – “The informational state I is realized at different levels of abstraction within the contexts of these LLMs”. What exactly is the informational state in this scenario? What are the axioms/criteria for determining whether a realization of the information state is good / bad? It would be good if authors can provide a deeper analysis on this.
8. Since the HAI-Co^2 framework is described at a high level, and from Section 5 it appears that its realization/implementation has various possibilities and is very non-trivial, it would be nice to see more than one example besides code generation, to further illustrate the usefulness of the proposed framework.

References of learning from noisy pairwise comparisons:

Shah, Nihar, et al. "Stochastically transitive models for pairwise comparisons: Statistical and computational issues." ICML (2016).

Mao, Cheng, et al. "Minimax rates and efficient algorithms for noisy sorting." ALT (2018).

---

> ### Author Response · Authors · 2024-11-08
> **Response to reviewer ucrj (Part I)**
>
> We thank the anonymous reviewer for their valuable comments. We have tried our best to revise the manuscript based on their suggested changes (marked in blue in the revised manuscript). In what follows, we explain the changes in the light of our rationale.
>
> ## Feasibility of learning from active human edits
>
> We have incorporated a discussion on the feasibility and methodological requirements to facilitate learning from active human participation under Section 4 in the revised manuscript. Precisely, we note that learning from active edits made by the expert requires the notion of neighborhood structure, either explicitly (with a well-defined notion of distance between two neighboring solutions such that one can compute the changes in utility as the user-made edit moves from one solution to another) or implicitly (by learning to map the changes to preferences). Prior research [1] has used the latter and shown that it is indeed a better way to incorporate preferences expressed via edits. However, their experimentation was on general-purpose text generation tasks. With expert domain problems where multiple utility dimensions are at play, an explicit structure of neighborhood can provide more transparent agent actions. Furthermore, in the example case study presented in Section 5, we now incorporate an option for the user to actively edit the solution.
>
> ## Multidimensional vs. scalar utility
>
> Please note that the aggregation of a multidimensional utility function into a scalar one is optional and our framework describes this as a design choice.
>
> ## Noisy binary preference
>
> Thank you for pointing out this issue. We have incorporated the mentioned literature under Section 4 (paragraph titled Multimodal human-AI interaction).
>
> ## Limitations and underspecification of the example case study
>
> We agree that the implementation of HAI-Co^2 is incomplete. Instead of an actual implementation of the underlying principles of HAI-Co^2 (e.g., neighborhood structure of the solution space, preference extraction from actual human participation, dedicated utility function tailored toward the expert problem, etc.), this implementation seeks to emulate the intended behavior using prompted LLMs, motivating the practicality of HAI-Co^2. We have reiterated this in the revised manuscript under Section 5. In the revised version, we have incorporated i) a Preference-decoder module that extracts and lists expert preferences that are used in the search across different levels of hierarchy, ii) the ability for the user to actively edit the solution, and for the AI to learn the preferences from such edits. We have further elaborated on how this implementation treats the informational state, and how the neighborhood structure can be interpreted using the stochastic decoding and edit distance between two solutions. Despite the incompleteness, a fundamental difference in terms of preference adherence can be observed. The GPT-driven search (which can be used in itself or in conjunction with human judgment) lets the UML generator explore the solution space while lessening the dependence on instruction following and selects the best course among many. Subsequently, it requires far less direct human engagement (by writing a prompt and asking the model to correct something) in this setup compared to vanilla LLMs.
>
> ## Examples from multiple domains
>
> Please note that our claims about the limitations of LLMs (or generative AI in general) in expert domain problems are founded in prior research that we have mentioned in the manuscript (such as planning [2], radiology [3], scientific writing [4]). We use the examples from code synthesis due to ease of visualization. In the revised manuscript, we propose a conceptual application of HAI-Co2 for another expert domain in Appendix A: for the problem of generation of a related work section.
>
> ## Quantitative evaluation of HAI-Co2
>
> We have implemented an offline human evaluation comparing the abilities of HAI-Co2 with that of the vanilla LLM (in this case, GPT-4 Turbo). Each evaluator was presented with a pair of user-AI interaction histories using the two methods and asked to compare them against different metrics. We have included the details and the results of the evaluation under Section 5 and Appendix B in the revised manuscript.
>
> Typo: Thank you for pointing it out. We have corrected the typo in the revised manuscript.

---

> > ### Author Response · Authors · 2024-11-08
> > **Part II**
> >
> > References:
> >
> > [1] Ge Gao, Alexey Taymanov, Eduardo Salinas, Paul Mineiro, and Dipendra Misra. Aligning LLM agents by learning latent preference from user edits, 2024
> >
> > [2] Subbarao Kambhampati, Karthik Valmeekam, Lin Guan, Kaya Stechly, Mudit Verma, Siddhant Bhambri, Lucas Saldyt, and Anil Murthy. LLMs Can’t Plan, But Can Help Planning in LLM-Modulo Frameworks, 2024.
> >
> > [3] Augustin Lecler, Loïc Duron, and Philippe Soyer. Revolutionizing radiology with GPT-based models: Current applications, future possibilities and limitations of ChatGPT. Diagnostic and Interventional Imaging, 104(6):269–274, 2023
> >
> > [4] Bassel Almarie, Paulo E P Teixeira, Kevin Pacheco-Barrios, Carlos Augusto Rossetti, and Felipe Fregni. Editorial - the use of large language models in science: Opportunities and challenges. Principles and Practice of Clinical Research, 9(1):1–4, Jul. 2023.

---

### Decision · Action_Editor_BCar · 2024-11-25

**Recommendation:** Reject

**Comment:**

This paper introduces HAI-Co², a framework for human-AI cooperative problem solving that addresses limitations in current LLM-human interactions. The framework focuses on expert domains where solutions are complex and preferences may be unclear initially. Three key innovations are proposed: (1) a hierarchical representation of solutions across multiple abstraction levels, allowing both high-level planning and detailed refinement, (2) a preference-based search methodology that incorporates both explicit and implicit human feedback, and (3) natural language as a central medium for human-AI coordination while supporting multiple modes of interaction.

While the proposed framework was appreciated by all reviewers, the lack of full specification of the frameworks and limited qualitative evaluation of the framework led the reviewers to not find the claims substantiated. While the authors put in a lot of work during the response period, the reviewers still felt that they failed to address their main concerns (they reviewers appreciated that the authors acknowledge the concerns). Here I list the main remaining concerns:
- Incomplete Specification: Critical components of the proposed framework are still left unspecified. Reviewer ucrj points out that *the utility function and the neighborhood structure still undefined in Section 5, and the corresponding principles that implementations of HAI-Co^2 need to satisfy (e.g., final solutions should maximize the scalar-valued utility; “The abstraction specification must adhere to the neighborhood structure on all levels of abstraction”) are still not examined.*
-  Implementation Gap: The paper propose a sophisticated framework with specific components however the actual implementation uses basic LLM prompting with core features (neighborhood structure, preference learning, abstraction mappings) unimplemented. This makes it hard to validate theoretical benefits.
- Weak Evaluation and Comparison to Existing Work: The current evaluation is small-scale (14 participants) and lacks qualitative metrics. Furthermore, there is no comparison to existing methods, and the evaluation is on an incomplete implementation. Thus the engagement with the existing literature remains superficial.

Given the lack of substantiation of claims, the paper is not ready to be published at TMLR. I encourage the authors to strengthen the submission, and resubmit by (1) implementing core framework features and providing robust evaluation and comparison to existing works, and (2) reframing their theoretical contribution with appropriately scoped claims and evidence. The reviewers have provided highly detailed and constructive reviews, and truly addressing their concerns would make this a strong submission.

**Audience:**

Yes, I believe the topic and content of the paper would be interesting to the TMLR audience.

**Claims And Evidence:**

Not entirely. While the authors acknowledged the concerns raised by the reviewers regarding the lack of substantiation of the claims and made significant effort to add details, the reviewers still feel that their main concerns have not been addressed. See comments for more details.

**Resubmission Of Major Revision:**

The authors may consider submitting a major revision at a later time.